# Segmentation Method of Cerebral Aneurysms Based on Entropy Selection Strategy

**DOI:** 10.3390/e24081062

**Published:** 2022-08-01

**Authors:** Tingting Li, Xingwei An, Yang Di, Jiaqian He, Shuang Liu, Dong Ming

**Affiliations:** 1Academy of Medical Engineering and Translational Medicine, Tianjin University, Tianjin 300110, China; ltt1999@tju.edu.cn (T.L.); dy_aquarius@tju.edu.cn (Y.D.); hhhhhjq@tju.edu.cn (J.H.); shuangliu@tju.edu.cn (S.L.); richardming@tju.edu.cn (D.M.); 2Tianjin Center for Brain Science, Tianjin 300110, China; 3Department of Biomedical Engineering, School of Precision Instruments and Optoelectronics Engineering, Tianjin University, Tianjin 300110, China

**Keywords:** segmentation, cerebral aneurysm, Transformer, 2D CNN, entropy

## Abstract

The segmentation of cerebral aneurysms is a challenging task because of their similar imaging features to blood vessels and the great imbalance between the foreground and background. However, the existing 2D segmentation methods do not make full use of 3D information and ignore the influence of global features. In this study, we propose an automatic solution for the segmentation of cerebral aneurysms. The proposed method relies on the 2D U-Net as the backbone and adds a Transformer block to capture remote information. Additionally, through the new entropy selection strategy, the network pays more attention to the indistinguishable blood vessels and aneurysms, so as to reduce the influence of class imbalance. In order to introduce global features, three continuous patches are taken as inputs, and a segmentation map corresponding to the central patch is generated. In the inference phase, using the proposed recombination strategy, the segmentation map was generated, and we verified the proposed method on the CADA dataset. We achieved a Dice coefficient (DSC) of 0.944, an IOU score of 0.941, recall of 0.946, an F2 score of 0.942, a mAP of 0.896 and a Hausdorff distance of 3.12 mm.

## 1. Introduction

Cerebral aneurysms occur in about 3% of the general population. With the development of neuroimaging, an increasing number of cerebral aneurysms are incidentally discovered [1]. A cerebral aneurysm is a pathological dilation of an intracranial blood vessel whose walls may be abnormally weak and prone to rupture. The rupture of aneurysms causes hemorrhage to the subarachnoid space surrounding the brain, and sometimes in the brain parenchyma [2]. Aneurysm size, shape and location are important factors of rupture [3]. Some traditional methods for cerebral aneurysms are based on statistical thresholding [4] and deformable models [5]. Linear convolution is applied to image processing [6]. The use of geometrically deformable models within a level-set framework is an automated segmentation technique for cerebral aneurysms, and these models’ ability to handle topological changes and adapt to complex structural shapes makes them well suited to automated segmentation of complex vascular structures [7]. These methods take lots of time and effort. Therefore, we need accurate and rapid automatic algorithms for the segmentation of aneurysms.

The development of artificial intelligence (AI)-based technologies in medicine is advancing rapidly, and AI has recently experienced an era of explosive growth across many industries—the healthcare industry is no exception. Research in multiple medical specialties has used AI to mimic the diagnostic capabilities of doctors [8,9,10,11]. Recent advances in deep learning [12] have made it possible to realize this idea. In this regard, convolutional neural networks (CNNs) [13] have been the most ground-breaking addition, which are dominating the field of computer vision. CNNs have also revolutionized semantic segmentation tasks. In medical image analysis, the novel CNN architecture is U-Net. Other authors have also built derivatives of the U-Net architecture [14]. U-Net comprises an encoder and a decoder. U-Net has shown impressive potential in segmenting medical images, even with a lack of labeled training data, to the extent that it has become the de facto standard in medical image segmentation [14]. Wasiq et al. proposed a coarse-to-fine method for locating pupils and eye center estimation by combining machine learning and image processing [15].

U-Net-based networks have become popular in medical image segmentation. MA-Unet [16] extracts multiscale features and combines local features with their corresponding global dependencies by attention mechanisms. Isensee et al. proposed nnU-Net, a deep learning framework that can automatically adjust the necessary relevant parameters according to the characteristics of the dataset [17]. Milletari et al. proposed a 3D variant of the U-Net architecture called V-Net, a fully convolutional neural network based on volumetrics [18]. Despite the inspiring results achieved, several issues exist in the developed approaches. For 2D networks, 2D inputs do not fully exploit the 3D image information [19,20,21]. However, 3D convolutions do not focus on the different in-plane and depth resolutions [21,22,23]. Therefore, in order to obtain the dependencies between channels, balancing between 2D and 3D is the key to further improving network performance. To solve this problem, H-DenseUNet [24] was proposed. It consists of a 2D DenseUNet for extracting intra-slice features and a 3D counterpart for hierarchically aggregating volumetric contexts for liver and tumor segmentation. Although it combines the advantages of 2D and 3D, it is not suitable for aneurysm segmentation with class imbalance, and separate modeling of intra-slice and inter-slice features will exacerbate class imbalance.

Attention mechanisms have recently become popular in computer vision. Instead of compressing the entire image or sequence into a static representation, attention allows the model to focus on the most relevant features as needed. Transformers are the focus of natural language processing. Although their impact has been limited in vision applications, an increasing number of methods with attention mechanisms are being proposed. Due to the limitations of convolutions, researchers tried to introduce Transformers in both the encoder and decoder. Xu et al. used LeViT as the encoder and passed the multiscale feature map to the decoder through skip connections, which achieved better performance in medical image segmentation [25]. Transformer networks in computer vision can be found in [26]. The Vision Transformer (ViT) [27] adapts Transformer models for computer vision applications.

Deep learning methods with good performance have recently been proposed to segment cerebral aneurysms [28,29]. Due to the class imbalance of medical images, the U-Net framework will cause false negative predictions. The lack of labeled medical images is also a big challenge. Feng et al. proposed a patch-based fully CNN architecture in retinal blood vessel segmentation tasks and used a patch selection based on entropy to ensure the retinal blood vessels were contained in the patches [30]. The entropy of images indicates the richness of information, where images with higher entropy will contain more foreground class objects. This approach will alleviate class imbalance.

To sum up, both 2D networks and 3D networks have their limitations. Although cascading two networks can achieve better results, it also increases the number of parameters and complexity of the network. In recent work, no novel network structure has been proposed for cerebral aneurysm segmentation, and no method to overcome class imbalance has been proposed for the characteristics of cerebral aneurysm data. Therefore, we improved the network structure according to the characteristics of the data and relieved the class imbalance of the data through the gradient entropy strategy.

This work was inspired by the successful application of Transformers in 3D CNNs in the field of brain tumor segmentation [31]. Due to the class imbalance of the CADA dataset, we introduce a patch-based architecture that relies on the 2D U-Net as the backbone and adds a Transformer block to capture remote information. We propose a patch selection strategy based on entropy to make the training data more sufficient. Then, three continuous patches are taken as inputs and a segmentation map corresponding to the central patch is generated.

As illustrated above, in this paper, the main contributions to aneurysm segmentation are as follows:(1)In order to obtain more sufficient training data, we used a new patch selection strategy. More training data with aneurysms will alleviate class imbalance.(2)We used three channels as inputs, which represents an approach between 2D and 3D. This approach can use 3D information and pay attention to the in-plane resolution.(3)We improved the recombination strategy. This will make the boundary of the segmentation target clearer.

The rest of this paper is organized as follows: Section 2 describes the proposed methodology in detail. Section 3 shows the experimental results. Finally, we discuss and conclude our paper in Section 4 and Section 5.

## 2. Materials and Methods

### 2.1. Dataset and Preprocessing

The MICCAI 2020 CADA challenge provided 109 cases. Image data of patients with cerebral aneurysms without vasospasm were collected for the purpose of assisting diagnosis and treatment [32]. The image data were acquired utilizing the digital subtraction AXIOM Artis C-arm system. Post-processing was performed using LEONARDO InSpace 3D (Siemens, Forchheim, Germany). After implementation of the contrast agent, a reconstruction of a volume of interest was selected by a neurosurgeon. The reconstructed images generally consist of 220 contiguous slices. The imaging parameters were as follows: in-plane size of 256×256; iso-voxel size of 0.5 mm. Patients were of different ages and genders, making the samples diverse.

#### 2.1.1. Slice Selection Strategy

In cerebral aneurysm images, approximately 98% of the pixels belong to the background, with the remaining 2% of pixels belonging to the foreground class. We selected the slices using the range entropy strategy proposed in [33]. For a given sample X∈ℝD×H×W, its spatial resolution is H×W, and its depth dimension is D (# of slices), normalizing the images using the following formula:(1)Xinorm=Xi−min(Xi)max(Xi)−min(Xi)∗255, i=[1,2,…,D]
where min(Xi) and max(Xi) denote the minimum and maximum values of the i-th slice in X. For every ten continuous slices, seven slices with the highest *RH* (range entropy) values were selected as the final slices annotated as XS∈ℝ7×D10×H×W. The calculation formula is as follows:(2)H(S)=−∑i=0255pilog2pi
(3)RH(S)=H(S)+w∗R(S)
(4)$$R(S)=\frac{1}{b}(\sum \max_{b}(S)-\sum \min_{b}(S))$$
where *H*(*S*) is the entropy of each slice, and $p_{i}$ is the probability of the pixel value i that exists in *S*; *R*(*S*) is a generalized range of the slice *S*, and *w* depicts the weight of the range *R*(*S*); $\max_{b}(S)$ denotes the top *b* maximum gray values in the slice *S*, and $\min_{b}(S)$ denotes the last *b* minimum gray values in the slice *S*, where *b* denotes the number of selected pixels [33].

#### 2.1.2. Sliding Window Strategy

In the cerebral aneurysm datasets, the number of available annotated images is not large enough for a good training model. Thus, for cerebral aneurysm segmentation, we used the patch method with the sliding window strategy.

From Figure 1a, under the same stride size, the proportion of patches containing aneurysms is large when the patch size is 96×96 pixel^2^. Although the proportions of 108×108 pixel^2^ and 128×128 pixel^2^ are larger, they will contain more noise due to the larger size. Additionally, in the segmentation of cerebral blood vessels [33], a patch size of 96×96 pixel^2^ was chosen, so we chose 96×96 pixel^2^ as the patch size.

The original resolution of the CADA dataset is 256×256 pixel^2^. From Figure 1b,c, although smaller strides produce more patches containing aneurysms, their performances are worse. This may be because the selected patches contain duplicate positive samples, resulting in information redundancy. When the stride is 32, the Dice is the best, and 32 is the largest common factor of 96 and 160. Considering the accuracy and computational complexity, we chose 32 as the moving stride of the sliding window. That is, a 96×96 pixel^2^ sliding window starts from the upper left corner of the slice with a moving stride of 32 pixels. Each step of moving the sliding window yields a corresponding patch. For a whole 256×256 pixel^2^ slice, we could acquire 36 patches.

#### 2.1.3. Patch Selection by Gradient Entropy Sampling

The generated patch is denoted as P∈ℝHp×Wp. Owing to the similarity between blood vessels and aneurysms, selecting patches only through information entropy will include a lot of noise. Since aneurysms have higher gradients than vessels, we combined information entropy with a gradient strategy for patch selection called *GH*, which is proposed as the gradient entropy strategy. The calculation formula is as follows:(5)GH(P)=H(P)+γ∗Gy(P)
(6)Gy(P)=1c∑maxc(P(i,j)−P(i,j−1))
[i=1,2,3,…,Hp, j=2,3,…,Wp+1], where *GH*(*P*) denotes the gradient entropy value of each patch, *P*(*i*,*j*) is the pixel value of the index (*i*,*j*) in *P*, Gy(P) represents the gradient in the y-direction and maxc(·) denotes the top *c* maximum Gy in the patch *P*. Patches with a higher *GH* are selected in the training set.

Applying the sliding window strategy, patch selection by information entropy sampling showed that the proportion of patches containing aneurysms was 12%., i.e., 12% of the selected patches included aneurysms. Additionally, through patch selection by gradient entropy sampling, the proportion was increased to 16%. The patches selected by the gradient entropy sampling strategy are shown in Figure 2. This shows that most of the selected patches were aneurysms or blood vessels.

### 2.2. Network Architecture

We chose the structure of U-Net as the backbone. Figure 3 presents the architecture of our network that consists of a CNN encoder block, a Transformer block and a decoder block with a shortcut connection at each resolution level. The encoder obtains high-dimensional features, and the decoder utilizes these encoded features to recover the segmentation target. The spatial attention is utilized to strengthen the region of interest on the feature maps while suppressing the potential background or irrelevant parts. Hence, we propose a Transformer block that shares space and learns the relation between these feature map embeddings using self-attention modules. The network uses the image information in three continuous patches to predict the segmentation for the center patch, and adjacent slices provide rich spatial information.

#### 2.2.1. Network Encoder

The encoder blocks are composed of ResNet34 and the Transformer block. ResNet34 is mainly composed of a Bottleneck. In order to prevent overfitting, we added a dropout block to the original Bottleneck (see Figure 4). ResNet34 is a type of neural network that captures more deeper features by using skip connections to “skip” a number of convolutional layers in every Bottleneck in the network. The structure of the ResNet34 encoder block is shown in Figure 5.

There are 4 layers in the ResNet34 encoder block, and the numbers of Bottlenecks in each layer are 3, 4, 6 and 3. The final output of ResNet34 can be written as F∈ℝC×H′×W′.

Next, we present a Transformer block comprising L repeated Transformer layers to achieve a global context using an attention mechanism. For a given F, it is flattened into a vector of size ℝd×N, by a linear projection operation Wp, resulting in f∈ℝd×N, d=512, N=3×3. c0=f+PE∈ℝd×N constitutes the input of the Transformer block, and PE is the learnable position embedding.

Each Transformer layer has a Multi-Head Attention (MHA) block and a feed-forward neural network (FFN), and the output of each layer can be calculated by the following formula:(7)$$c_{i}^{\prime }=MHA(LN(c_{i-1}))+c_{i-1}$$
(8)$$c_{i}=FFN(LN(c_{i}^{\prime }))+c_{i}^{\prime }$$
where LN(∗) denotes the layer normalization, and $c_{i}$ is the output of the ith (i∈[1,2,…,L]) Transformer layer.

#### 2.2.2. Network Decoder

To fit the input dimension of the 2D CNN decoder, f is then reshaped to f′∈ℝd×H″×W″ by a feature mapping module. The decoding process corresponds to the encoding process, which combines local and global features until the original resolution is restored and pays more attention to the local context to obtain edge and semantic information. Additionally, through cascaded upsampling operations and convolution blocks, the final segmentation map S∈ℝHp×Wp is generated.

### 2.3. Prediction

After the training phase, the trained model can only segment 96×96 images. In the prediction phase (see Figure 6), the input samples were segmented into 96×96 patches using the sliding window strategy. Then, the trained model generated segmentations of patches. The segmentation maps were recombined to obtain the final segmentation map. The stride was 32, and adjacent patches had overlapping pixels. For each pixel, the average strategy [33] is where the aneurysm probability of each pixel is obtained by averaging possibilities over all the predicted patches covering the pixel. This solution will lose a lot of detail and is time-consuming. We propose a new recombination strategy.

For each image, each row of the sliding window produced six overlapping patches, and a total of 36 patches were generated. There were C42+1=7 cases for recombining each row. Similarly, each column also had 7 cases. There was a total of 6×74+73=14,749 recombination situations. Due to the robustness of the model, we randomly selected only 49 of these cases, and then 49 segmentation maps were obtained. We averaged these 49 maps to obtain the SAver∈ℝH×W; the SAver∈ℝH×W had obvious clipped parts. We added the self-attention of each patch to the segmentation result Scandidate∈ℝ36×H×W to ensure the integrity of the segmented aneurysms by replacing the corresponding position of SAver with each patch segmentation map. If SAver contained aneurysms and the number of aneurysms contained in Scandidate was more than the threshold η, and if SAver did not contain an aneurysm, but the number of aneurysms contained in Scandidate was more than the threshold 36−η, then we selected the one with the largest aneurysm size among the 36 results as the final segmentation result. Otherwise, we considered the image to be without an aneurysm.

## 3. Experiment

### 3.1. Data Augmentation

Most selected patches contained aneurysms or vessels. For data augmentation, the following data augmentation techniques were applied: (1) random cropping; (2) horizontal flipping; (3) 45° rotation. We only used data augmentation on the training data. After applying patch selection, each sample X∈ℝD×H×W was cut into 200 inputs consisting of three consecutive patches. In terms of the specific process of random clipping, for every three consecutive slices, we chose an identical random position to crop, generating three consecutive patches, with a total of D//3 inputs. Then, we applied horizontal flipping and 45° rotation on patches generated by random clipping. To sum up, after data augmentation, D additional inputs were generated.

### 3.2. Evaluation Metrics

The metrics of evaluation were the Dice score, recall, Hausdorff distance (95%), F2 score, mean average precision (mAP) and Intersection over Union (IOU) score.

*Dice similarity coefficient*: The Dice similarity coefficient (Dice) [34] is a metric used for assessing the quality of segmentation. It measures the similarity between the predicted label and ground truth.

$Dice=\frac{2\left|Pre\cap G\right|}{\left|Pre\right|+\left|G\right|}$, where *Pre* is the predicted segmentation map, and *G* is the ground truth.

*Hausdorff distance*: A high Hausdorff distance value implies that the two contours do not closely match. It is a symmetric measure of distance between two contours and is defined as [35]
$$H(Pre,G)=\max(h(Pre,G),h(G,Pre)),h(Pre,G)=\max_{p_{i}\in Pre}\min_{g_{i}\in G}\left\|p_{i}-g_{i}\right\|$$

*Recall*: Recall is the ability to segment the region of interest in the segmentation experiment. It indicates the proportion of all true positives that are correctly predicted.

$recall=\frac{TP}{TP+FN}$, where *TP* means true positive predictions, and *FN* means false negative predictions.

*F2 score*: F2 = $5\cdot \frac{precision\cdot recall}{4\cdot precision+recall}$, where $precision=\frac{TP}{TP+FP}$.

*mAP*: The mAP is the average precision of all categories detected, that is, the average precision of segmenting the foreground and background.

*IOU*: $IOU=\frac{Pre\cap G}{Pre\cup G}$, the closer the *IOU* is to 1, the better the segmentation result.

### 3.3. Implementation Details

We used a random split (70% training, 10% validation and 20% test) at the patient level and conducted a five-fold cross-validation evaluation. All experiments were implemented in Pytorch.

In the data preprocessing stage, the parameters should be determined. w in Equation (3), *b* in Equation (4), γ in Equation (5) and *c* in Equation (6) were set to 0.05, 10, 0.1 and 20, respectively. Because the patch size is 96×96 pixel^2^ and the stride of the sliding window is 32, a patch can be composed of 1/3 of three adjacent patches. η in Section 2.3 was set to 36 − 3 = 33. The dropout regularization with *p* = 0.2 was used.

We used Resnet34 pretrained on ImageNet as the CNN encoder block. We set the training of our model on Pytorch with an initial learning rate of $1\times 10^{-4}$. If the Dice score on the validation dataset did not improve for 15 epochs, the learning rate was reduced to half of the original rate. It was optimized by Adam with a batch size of 32. The training iterated 50 epochs on a single NVIDIA GeForce RTX 3090 GPU with 24 GB memory. The softmax Dice loss was employed to train the network.

To demonstrate the advantages of our work, we compared it with other methods (3D U-Net, DeepLabV3+, DeepLabV3, Linknet, FPN, UNet++). (1) The 3D U-Net took a learning rate of $1\times 10^{-4}$, and it was optimized by Adam and trained with an NVIDIA GeForce RTX 3090 for 500 epochs from scratch using a batch size of 4. (2) DeepLabV3+ was pretrained on ImageNet, and then it was trained on our dataset with an initial learning rate of $1\times 10^{-4}$. It was optimized by Adam with a batch size of 16. (3) DeepLabV3 implemented the same setup as DeepLabV3+. (4) Linknet was pretrained on ImageNet. Its initial learning rate was $1\times 10^{-5}$, it was optimized by Adam and the weight decay was $1\times 10^{-4}$. It was trained with an NVIDIA GeForce RTX 3090 for 50 epochs using a batch size of 16. (5) FPN implemented the same setup as Linknet. (6) UNet++ implemented the same setup as Linknet.

### 3.4. Segmentation Result and Comparisons

We conducted a five-fold cross-validation evaluation on the training set, and our method achieved an average Dice score of 0.944, an IOU score of 0.941, recall of 0.946, an F2 score of 0.942, a mAP of 0.896 and a Hausdorff distance of 3.12mm, which are comparable or higher results than those of previous state-of-the-art (SOTA) methods presented in Table 1. Compared with the 3D U-Net, our method showed superiority in four metrics, with a significant improvement.

Figure 7 shows example slices from test images in the dataset and the segmentations predicted by the proposed method. We observed that the proposed method could accurately segment fine or large cerebral aneurysms.

Figure 8 shows that the predicted segmentations of DeepLabV3+ mostly contained holes for larger aneurysms, while for smaller aneurysms, it was prone to false negative predictions.

### 3.5. Ablation Study

We designed different ablation studies to evaluate the contribution of the gradient entropy sampling strategy and the three-channel input, based on the patch and recombination strategies.

In the data preprocessing, the gradient entropy sampling strategy was implemented to generate a sufficient number of training patches from the limited images. We compared the traditional entropy sampling strategy with our strategy, and the result is shown in Table 2. The visual result is shown in Figure 9. The last row shows the segmentation of our method, where the segmentation is almost the same as the ground truth. As shown in Figure 9, our gradient entropy sampling strategy can better distinguish vessels and aneurysms. It is easy for information entropy sampling selection to produce false positives, and it has poor segmentation performance concerning smaller aneurysms.

As shown in Table 2, our model implementing the proposed gradient entropy sampling strategy achieved a better segmentation performance than the traditional entropy sampling strategy. This demonstrates that the quality of the training dataset can also improve the performance of segmentation, and that the gradient entropy sampling strategy provided more patches that contained aneurysms.

Table 3 shows that our recombination strategy achieved a better performance than the average strategy. Our recombination strategy combines the global attention and self-attention of patches, which not only ensures the overall performance of segmentation but also ensures the integrity of the foreground.

A characteristic of cerebral aneurysm image data is class imbalance. Furthermore, the patch-based approach ensures that the classes of the input data are balanced and the patches give the network access to local information about the pixels, which has an impact on the overall prediction. Table 4 shows the ablation experimental results of one patch as an input, three continuous slices as an input and our proposed three continuous patches as an input. The three continuous patches as an input achieved the best results, the three continuous slices as an input achieved the worst Hausdorff distance and the one patch as an input achieved the worst result for the other three metrics. This indicates that the multichannel input provided more 3D spatial information. The Hausdorff distance implies the degree to which two contours closely match. The three continuous slices as an input lost detail of the local information, resulting in a poor effect of aneurysm contours.

## 4. Discussion

In the experiment, we found that the proposed segmentation network architecture had a great advantage over the previous algorithms. In the ablation study, we verified the validity of the gradient entropy selection strategy, and the result shows that it had a better performance in aneurysm segmentation than traditional information entropy selection. This provides ideas for the development of small medical area segmentation fields. It reduces false positive and false negative predictions. Three channels provided more 3D information and had a significant effect on extracting features. The results in Table 1 show that our network was optimal in all six metrics. Additionally, we can see from Figure 8 that, whether it is a large aneurysm or a small aneurysm, our model’s performance was better. Due to the size of the training set being small, the segmentation performance of the 3D U-Net was not ideal. Compared with the remaining 2D SOTA segmentation networks, our input provided additional 3D information, so the segmentation results were better than those of other networks. Although the DeepLabv3+ network is better than the remaining networks, there are still holes in the segmentation of aneurysms of a large size.

For our method, although the average performance was better than the other models, there were still a small number of samples whose segmentation was not ideal, which may be due to the excellent threshold selection when selecting patches. As a result, the network did not learn the information specific to the sample, and thus the hyperparameter selection should be improved.

In the clinical field, aneurysm screening is essential, as early detection can prevent stroke. Relying on a doctor’s manual observation is inefficient, and different doctors have different evaluation criteria. In this paper, we provided a new efficient aneurysm segmentation algorithm, which facilitates rapid diagnosis and unified evaluation criteria.

## 5. Conclusions

Cerebral aneurysm is one of the most common cerebrovascular diseases, and its rupture has a high mortality rate from subarachnoid hemorrhage (SAH). Due to the limited training data, existing automatic segmentation methods cannot segment aneurysms sufficiently, so we adopted an entropy selection strategy to provide informative training data. Specifically, we proposed a patch-based segmentation model. Compared with full resolution inputs, the selected patches are only part of the training data, thus preventing overfitting. We used the gradient entropy strategy to select patches that may contain aneurysms, improving and speeding up the network. The experimental results show that better training data can also improve the network’s performance. We proposed a new application scenario for entropy. In future work, we will continue to design deep architectures for small datasets in medical image processing.

## Figures and Tables

**Figure 1 entropy-24-01062-f001:**
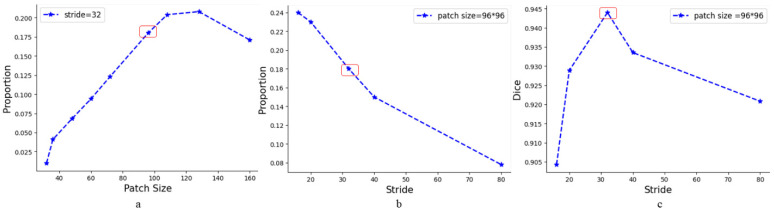
The choice of patch size and stride size. (**a**) The proportion of patches containing aneurysms with different patch sizes when the stride size is 32. (**b**) The proportion of patches containing aneurysms with different stride sizes when the patch size is 96×96 pixel^2^. (**c**) The Dice at different stride sizes.

**Figure 2 entropy-24-01062-f002:**
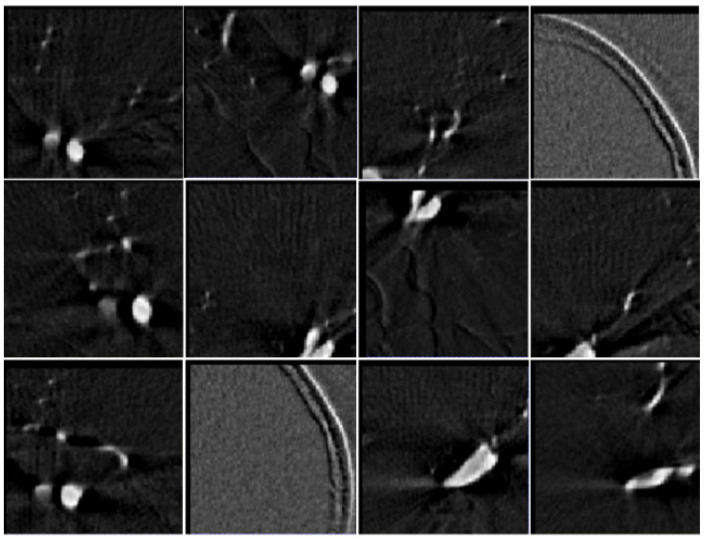
Selected patches based on the gradient entropy sampling strategy.

**Figure 3 entropy-24-01062-f003:**
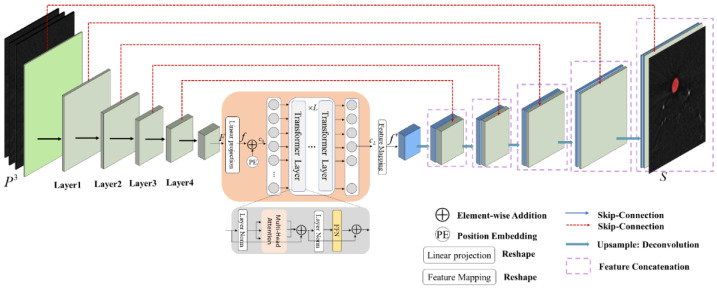
The overall network. Three consecutive patches are used as inputs. The left part of the network is the encoder based on ResNet34, and each green block corresponds to the layer of ResNet34. The middle part of the network is the Transformer block. The right part of the network is the decoder, and each blue block corresponds to the upsampling block.

**Figure 4 entropy-24-01062-f004:**
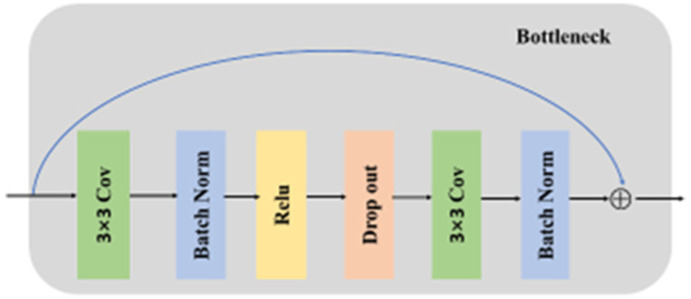
The visual of the Bottleneck.

**Figure 5 entropy-24-01062-f005:**
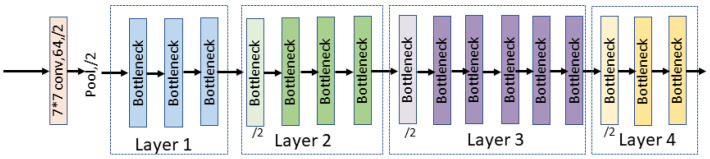
The structure of the ResNet34 encoder block.

**Figure 6 entropy-24-01062-f006:**
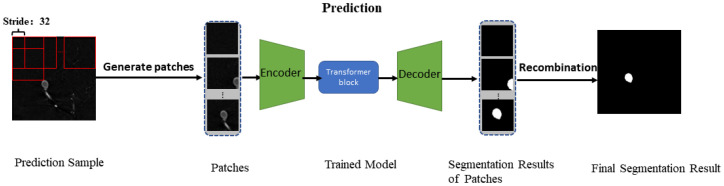
The pipeline of prediction. The test set samples first generate patches through the sliding window, then input them into the trained network to generate the corresponding prediction and finally splice the final results.

**Figure 7 entropy-24-01062-f007:**
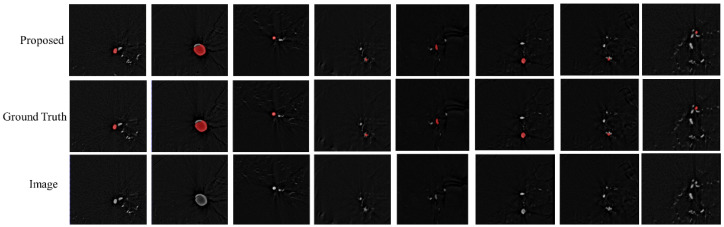
The visual segmentation result of our method.

**Figure 8 entropy-24-01062-f008:**
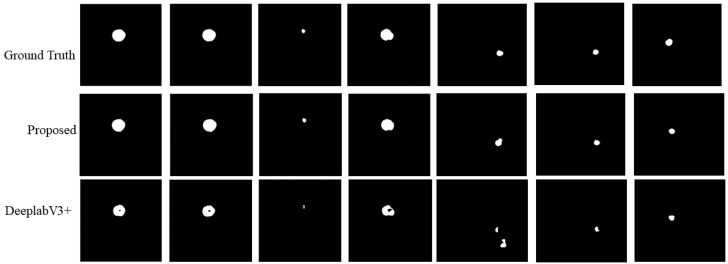
The visual results of our method and DeepLabV3+.

**Figure 9 entropy-24-01062-f009:**
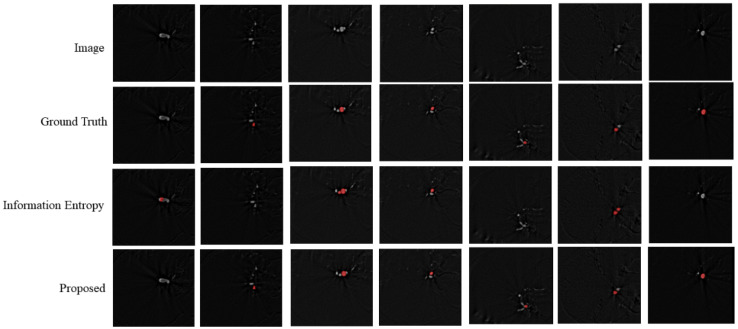
The visual results of different patch selection strategies.

**Table 1 entropy-24-01062-t001:** Comparison of SOTA methods.

Model	Dice	IOU	Recall	Hausdorff_95	mAP	F2 Score
3D U-Net	0.631	0.521	0.690	19.1	0.857	0.653
Linknet	0.867	0.856	0.952	19.85	0.893	0.859
DeepLabV3	0.916	0.912	0.936	10.22	0.632	0.897
FPN	0.929	0.925	0.925	8.40	0.838	0.936
DeepLabV3+	0.937	0.934	0.939	6.36	0.835	0.936
UNet++	0.939	0.935	0.945	10.28	0.874	0.937
**Proposed**	**0.944**	**0.941**	**0.946**	**3.12**	**0.896**	**0.942**

**Table 2 entropy-24-01062-t002:** Ablation study on the gradient entropy sampling strategy.

Model	Dice	IOU	Recall	Hausdorff_95
Information entropy	0.928	0.925	0.945	4.42
**Proposed**	**0.944**	**0.941**	**0.946**	**3.12**

**Table 3 entropy-24-01062-t003:** Ablation study on the recombination strategy.

Model	Dice	IOU	Recall	Hausdorff_95
Ours w/o post	0.942	0.938	0.942	4.00
**Proposed**	**0.944**	**0.941**	**0.946**	**3.12**

**Table 4 entropy-24-01062-t004:** Ablation study on the three-channel input and resolution.

Model	Dice	IOU	Recall	Hausdorff_95
One-patch input	0.931	0.919	0.932	6.40
Based on slices	0.939	0.934	0.946	7.42
**Proposed**	**0.944**	**0.941**	**0.946**	**3.12**

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
