# Peer review of "Segmentation Method of Cerebral Aneurysms Based on Entropy Selection Strategy"

_entropy, 2022, doi:10.3390/e24081062_

Round 1

Reviewer 1 Report

Improve the Academic/English writing style throughout the manuscript. 

Use abbreviations consistently throughout the paper. See line 43 e.g., AI or artificial Intelligence?

Likewise, spelling mistakes such as line 42-43 etc., needed to be corrected. 

Line 80 Page 2: 'Due to the class imbalance of medical images, U-Net 80

framework will cause false-negative predictions.'

Why authors resolve this matter through techniques and not data balancing? E.g., you could gather more data or use DL or other data augmentation

methods to balance the dataset.

I would recommend to add the following recent works related to image segmentation:https://www.mdpi.com/1424-8220/20/13/3785; https://ieeexplore.ieee.org/abstract/document/7737454

Before presenting the contribution of the work, I would recommend to further highlight the limitations in existing related works.

Dataset: Can you provide further details about the dataset? How it is annotated? Proportion used for train/test etc?

What is the diversity within the data? e.g., environment, lightening conditions, patients' diversity etc.

2.1.2. Why and how did you pick the patch size? if it is through experiments, better to show/add outcomes in Appendix or in this section.

Figure 2 caption: Captions must consist of detailed information about the figures. 

Experiments: You mentioned data augmentation is used. Firstly, what was the size of dataset after augmentation?

How the augmentation is used? For example did you use on Training or testing or both. need to clarify this.

Also, mAP is the standard outcome from such DL model along with IoU and Accuracy. 

Discussions: Better to compare your outcomes with existing state-of-the art and enrich the discussions with critical analysis of your

outcomes and existing works. 

Highlight some limitations in your work in this section.

Reviewer 2 Report

The authors propose a 2D U-net backbone and transformer block machine learning network for segmentation of cerebral aneurysms in medical images. Gradient-enropy sampling was introduced to generate training patches for training data selection.

The overall paper is nicely written and sound.

I would like to suggest recommendations

1. Figure 2 caption can be extended to briefly describe the network design .

2. The original CADA challenges uses extensively the F2 score for competition. How would the results from authors network compare in F2 score in comparison to 3D Unet deeplab v3 and deeplabv3+?

Reviewer 3 Report

The present research propose an automatic solution for the segmentation of
cerebral aneurysms using a 2D U-NET architecture and adding a Transformer to capture remote information. The proposed method is compared using the CADA dataset provided by MICCAI-2020 with different methods of the state of the art. The paper is well written in English and it presents an informative state of the art, which is pertinent to introduce the problem.

My main concerns with the present research are the following:

a) In line 52, Please improve  "U-Net and U-Net like models.."
b) In line 95, the patch selection strategy to alleviate class imbalance cannot be proposed, since it is a common strategy in deep learning for image segmentation.
c) How were the size and stride of patches determined?, Please present an analysis of different configurations.
d) Please improve the quality of Figure 5.
e) In data augmentation, please explain the use of random cropping, horizontal flipping and rotation 45.
f) In Table 1,2,3,4, Please use "Proposed" instead "Ours".
g) In Table 1, the comparison with only 3 SOTA methods is poor, please increase the number of SOTA methods in the comparison. The presented results need to be improved.
h) Please increase the number of samples in Figures 6,7,8.

Round 2

Reviewer 3 Report

My comments were properly addressed.